# The imprint of erosion by glacial lake outburst floods in the topography of central Himalayan rivers

Maxwell P. Dahlquist[1] and A. Joshua West[2]

[1]Department of Earth and Environmental Systems, Sewanee: The University of the South, Sewanee, Tennessee, 37383, USA
[2]Department of Earth Sciences, University of Southern California, Los Angeles, California 90089, USA

**Correspondence:** Maxwell P. Dahlquist (mpdahlqu@sewanee.edu)

**Abstract.** In steep landscapes, river incision sets the pace of landscape evolution. Transport of coarse sediment controls incision by evacuating material delivered to river channels by landslides. However, large landslide-derived boulders that impede bedrock erosion are immobile even in major runoff-driven floods. Glacial lake outburst floods (GLOFs) mobilize these boulders and drive incision, yet their role in regional-scale erosion is poorly understood, largely because of their rarity. Here, we find a topographic signature consistent with widespread GLOF erosion in the Nepal Himalaya. Our interpretations emerge from analysis of normalized channel steepness patterns, knickpoint distributions, and valley wideness. In rivers with glaciated headwaters that generate GLOFs, valleys stay narrow and relatively free of sediment, with bedrock often exposed to erosion. In turn, tributaries to these valleys are steep, allowing less efficient erosional regimes to keep pace with GLOF-driven incision. Where GLOFs are less frequent, valleys are more alluviated and incision stalls. Our results suggest the extent of headwater glaciation may play an important role in erosion of Himalayan river valleys and deserves more attention in future work.

## 1 Introduction

### 1.1 Motivation

The erosion of mountainous topography crafts the shape of Earth's surface, influences atmospheric circulation and global climate, modulates global carbon and nutrient fluxes, and affects the tempo of natural hazards including earthquakes and landslides (Raymo and Ruddiman, 1992; Hilton and West, 2020; Steer et al., 2014; Larsen and Montgomery, 2012). At elevations above the equilibrium line altitude (ELA), snow persists from one year to the next, forming glaciers that carve textbook U-shaped valleys (Davis, 1900). Fierce debates have centred on the notion that a "glacial erosion buzz-saw" limits the total height and relief of mountain ranges (Brozović et al., 1997; Egholm et al., 2009; Thomson et al., 2010; Cunningham et al., 2019) but even the proponents of this idea generally assume that the influence of glacial erosion fades below the ELA (Prasicek et al., 2018).

Many studies have noted the dramatic erosive power of GLOFs, which arise from the sudden and catastrophic draining of ice or moraine dammed lakes (Mason, 1929; Haeberli, 1983; Montgomery et al., 2004). The resulting floods can scour river valleys for 10s to 100s of kilometres downstream (Cenderelli and Wohl, 2003; Baynes et al., 2015; Jacquet et al., 2017; Lang et al., 2013; Cook et al., 2018), in some cases mobilizing boulders that otherwise remain stationary even during heavy rainfall-driven

flooding (Cook et al., 2018; Xu, 1988; Huber et al., 2020). The leading edge of an outburst flood remains below its transport capacity because the velocity of the water bore exceeds that of entrained bedload. Mobilization of sediment requires flow to exceed a certain critical shear stress, which is a function of the average particle size and the fluid and sediment densities. The bed shear stress exerted by flow is dependent on the flow depth and velocity, but energy expended by bedload transport decreases the ability of a flood to mobilize new material (Shields, 1936). Since GLOFs maintain a relatively sediment-sparse pulse of water at their front, they remain capable of mobilizing additional material as they progress downstream. These features make GLOFs highly effective incision mechanisms even in low-gradient channels (Cook et al., 2018; Pickering et al., 2019). These events can thus extend the imprint of glacier-associated erosion well below the elevations that support glaciers themselves.

While the dramatic effects of GLOFs have been well-documented, their rarity has made it challenging to identify whether these floods are sufficiently frequent and widespread to play an important role in controlling the long-term evolution of mountain topography. Evidence from glacial lake-derived valley fill and river profiles in the Shyok and Indus valleys suggests that fluvioglacial interactions promote incision into the western edge of the Tibetan Plateau (Scherler et al., 2014). Yet this effect is juxtaposed against the long-term inhibition of erosion as a result of lakes formed by glacial dams (Korup et al., 2010). Here, we evaluate the valley and channel morphology of rivers draining the Nepal Himalaya, finding signatures that are consistent with a systematic role for GLOFs as important agents of long-term erosion. Specifically, we compared rivers that have glaciated (or recently glaciated) headwaters versus those that do not, finding that rivers with glaciated headwaters are distinct both in valley width and channel steepness of tributaries to the main glaciated trunk streams draining the central Himalaya. Furthermore, we observe that knickpoints are concentrated in tributaries more likely to have experienced repeated GLOFs. We attribute these differences to the long-term erosional imprint of repeated GLOFs. Our results suggest "top-down" glacially conditioned erosion may be important across more of the landscape in major mountain ranges than currently recognized, which would have fundamental implications for the coupling of tectonics, erosion, and landscape evolution, and for the interpretation of tectonic processes from river channel form.

## 1.2  Setting: The role of GLOF erosion in the Nepal Himalaya

The Nepal Himalaya are a leading exemplar of an actively eroding mountain range, offering an ideal environment for investigating the relationships among tectonics, topography, and erosion. The major rivers in Nepal have their headwaters in Tibet and flow across the High Himalaya and Middle Hills, crossing a sharp physiographic transition (PT) as they pass into the lower relief zones of the Himalayan Middle Hills, and then transiting through the Siwalik Hills before ultimately draining onto the Gangetic Plain (Figure 1A). Tributaries to these rivers drain widely varying topography characterized by diverse geomorphic processes (Whipple and Tucker, 1999; Montgomery and Foufoula-Georgiou, 1993). Many of the major rivers have large areas of glaciated headwaters, and much attention has focused on the hazard posed to this region by increasing GLOF frequency in a warming climate (Korup and Tweed, 2007; Veh et al., 2020). Investigation of the role of GLOFs in shaping this landscape remains limited largely to individual case studies (Cenderelli and Wohl, 2003; Cook et al., 2018), along with identifying sedimentary evidence of past GLOF activity (Pickering et al., 2019; Huber et al., 2020). To test for a signature of pervasive GLOF control on erosion across the central Nepal Himalaya, we calculated metrics of river profile morphology, specifically (1)

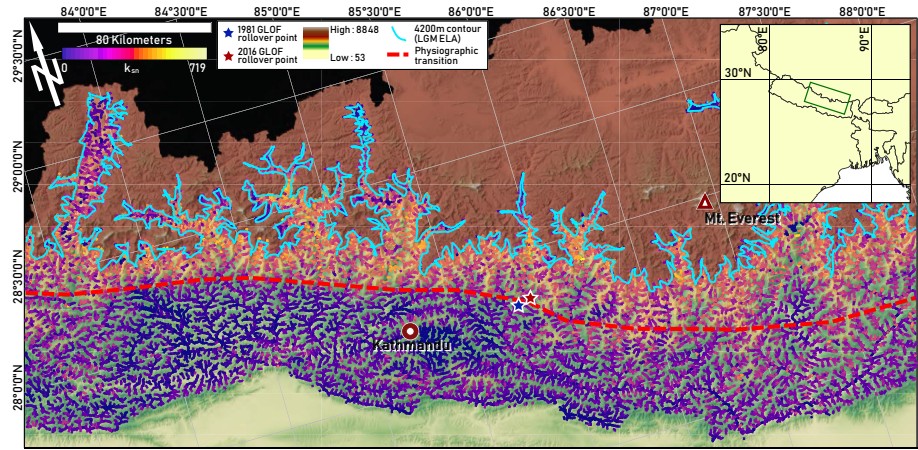

**Figure 1.** (a) Overview map of the study area, showing equilibrium line altitude at the Last Glacial Maximum (LGM ELA) along with other points of interest. $k_{sn}$ values are overlain on river network for elevations below LGM ELA (Asahi, 2010) and were analysed only where direct glacial action did not appear to be a major erosion mechanism. PT is assessed visually based on the location of abrupt change in relief.

normalized channel steepness adjusted for precipitation and evapotranspiration, (2) the prevalence of knickpoints in tributary channels, and (3) valley width and normalized valley wideness. We interpret the river channel metrics in the context of the upstream drainage area above the last glacial maximum ELA (LGM ELA), estimated to have been 4200 meters in the Nepal Himalaya (Asahi, 2010). We assume that the frequency of GLOFs was proportional to the potentially glaciated terrain in each basin. Note that in all our analyses, for simplicity we refer to rivers with parts of their watersheds above 4200 meters as having glaciated headwaters.

We used the LGM ELA on the basis that river morphology expressed today reflects the integration of erosional processes over the several thousand years of glacial retreat (Ray and Srivastava, 2010). There are several important caveats to this assumption. First, outburst floods originating from landslide-dammed lakes are also common in the Himalaya and are also important geomorphic agents (Hewitt, 1998), but we do not expect a straightforward relationship between upstream glaciers and landslide-dammed lakes, so our analysis based on drainage area above the LGM ELA limits our focus to the particular effect of GLOFs. The stochastic distribution of landslides and landslide dams makes spatial analysis of the topographic effect of landslide lake outburst floods more difficult, though an interesting problem for future work. Second, the assumption that drainage area above the ELA is proportional to GLOF frequency is imperfect, since, for example, the extent of glaciation on the Tibetan Plateau during the LGM is debated even though this area lies above the ELA (Kirchner et al., 2011). Third, factors such as valley geometry, glacial dynamics, and seismicity also play roles in GLOF generation, though evaluating these factors is beyond the scope of this study. Fourth, upstream glaciers also have an important influence on non-outburst flood runoff, contributing meltwater during the hottest months of the summer, generally during the monsoon months. This meltwater can exacerbate monsoon-driven flooding even in the absence of GLOF generation (Lutz et al., 2014). We reduced the likelihood of region-to-region variability in GLOF frequency affecting our results by focusing our study area within the Central Himalaya

region which is frequently considered as a coherent unit in hazard analyses of GLOFs (Veh et al., 2019; Fischer et al., 2021). In this region, total glacier mass balance, i.e. the regional difference between ice accumulation and meltoff, which is intrinsically tied to glacial volume, has been found to be related to the frequency of floods originating in moraine-dammed lakes (Fischer et al., 2021). The relationship between upstream drainage area above the ELA and outburst flood frequency is likely non-linear, but we maintain that it is a reasonable proxy for regional-scale assessment. Despite the many complicating variables at play which we do not attempt to entirely account for here, we will test the hypothesis that the immobility of large boulders in monsoon-driven floods points to GLOFs as an important erosional mechanism. To do this, we present a conceptual model for a potential river morphologic response if repeated GLOFs are indeed effective enough as geomorphic agents to leave topographic evidence in the regional landscape. We then test this model against several lines of topographic evidence found in the Himalayan landscape.

## 1.3 Conceptual model for river morphologic response to GLOF erosion

At elevations below the extent of glaciation, rivers are the main pacemakers of erosion. The erosive power of rivers is controlled by their base level, which is the lowest elevation of active fluvial erosion. Base level is scale-dependent, and might be defined for a tributary as the elevation of the junction with a higher-order stream, affected by incision and aggradation in the trunk stream. Regionally, it might be the defined by elevation of an alluvial fan at a range front, while globally, base level is sea level. Uplift of mountainous terrain effectively decreases regional base level, driving rivers to steepen and incise more deeply into uplifting rock. This incision steepens surrounding hillslopes, which respond by eroding faster (Burbank et al., 2003). According to the detachment-limited framework for river evolution, pulses of fluvial erosion are driven by base level changes that begin at low elevations (e.g., at river outlets) and propagate upstream along the mainstem of a river and into its tributaries, producing a wave of incision and hillslope lowering that works its way through the landscape (Figure 2A-C) (Howard, 1994). This simple conceptual model finds natural expression in fault-block mountains where uplift is focused on a single fault at the base of the range (Whittaker, 2012). In such settings and under the right conditions, the topographic profiles of rivers preserve quantitative information about the tectonic and geodynamic drivers of uplift, or about past change in climate (Whipple and Tucker, 1999). In more complex mountain ranges, numerous other processes can affect river incision and erosion, including differential rock uplift associated with multiple active tectonic features (Kirby and Whipple, 2001), gradients in precipitation and channel width (Roe et al., 2003; Finnegan et al., 2005), and variations in lithology, rock strength, and sediment availability (Sklar and Dietrich, 2001, 2006). In addition, high-magnitude, infrequent events, such as GLOFs, play key roles in erosion (Kirchner et al., 2001; Cook et al., 2018) – yet the role of outburst floods in particular in modulating the response of incision to uplift is poorly understood.

## 1.4 Morphometric proxies of GLOF erosion

We test for three predicted effects of GLOF-driven erosion on the topographic form of rivers in the central Himalaya. The first of these is the steepness of river channels. Normalized channel steepness ($k_{sn}$) represents the steepness of channels after accounting for the typically concave form of most river profiles. This concave form is reflected in a power law relationship

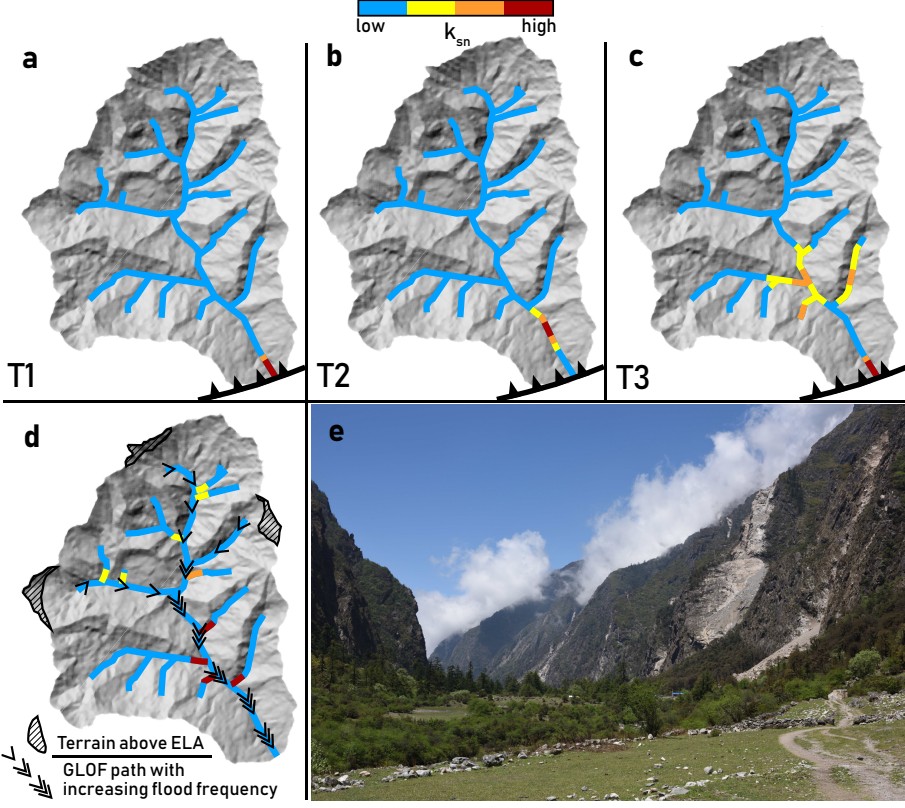

**Figure 2.** (a) Schematic of predicted $k_{sn}$ patterns arising from erosion driven by upstream knickpoint migration resulting from base level fall, including knickpoint diffusion described in alluvial and bedrock-alluvial channels (Rosenbloom and Anderson, 1994). (a-c) represent time steps showing the evolution of $k_{sn}$ patterns following a base level fall initiating at the thrust fault at the outlet of the catchment (panels A-C reflect temporal progression). In 1c, a second base level fall has initiated. (d) Schematic of $k_{sn}$ patterns we hypothesize to arise from erosion driven by GLOFs originating from the high-elevation regions shown as terrain above ELA. Steepened reaches formed near confluences may migrate upstream, steepening the basin at large. Our aim in illustrating the simple scenario shown in panels a-c is not to suggest it as a plausible representation of the tectonic geomorphology of the Himalaya, but instead to contrast the end-member expectations from erosion purely driven by changes in base level versus the conceptual model we propose for glacial lake outburst flood (GLOF)-driven erosion, in panel d — while recognizing that actual erosion in Himalayan river valleys will involve a collaboration between these end-member scenarios. (e) Photograph from Langtang Valley, Nepal, showing steep inner valley walls and steep tributary catchments entering the trunk valley 1 kilometer below the lowest identified glacial surfaces. Photo location is 28.200° N, 85.460° E. Photo courtesy of William Medwedeff.

between channel slope ($S$) and upstream area ($A$), where

$$S = k_s A^{-\theta} \tag{1}$$

If $\theta$ is fixed to a best-fit reference value, the normalized channel steepness $k_{sn}$ provides a basis for comparing the relative
steepness of different channels (see Methods) (Flint, 1974). Differences in $k_{sn}$ between river segments have been attributed to

variations in uplift (faster uplift requires a steeper, more energetic river for incision to keep pace), local rock strength (stronger rocks require more energy to erode), sediment supply (competing effects of tools and cover either enhance or inhibit erosion), and climate (less discharge means less erosive power, requiring steeper channels). Importantly for our purposes, GLOFs may influence $k_{sn}$ because they are highly effective erosional agents, capable of exerting extreme bed shear stresses by the progress of a high-velocity pulse of water even in a low-gradient river. High-magnitude, low-frequency discharge events, of which lake outburst floods are the apotheosis, are recognized as a critical control on erosion and on the geometry of channels, particularly where discharge thresholds for initiation of erosion are high (Snyder et al., 2003; Lague et al., 2005; Turowski et al., 2009; DiBiase and Whipple, 2011). As a result, erosional efficiency can be enhanced under conditions where channel steepness is low, mean discharge and discharge variability are high, and incision thresholds are high (DiBiase and Whipple, 2011). The major rivers of the Nepal Himalaya should meet these conditions, with discharge peaks defined by catastrophic outburst floods and incision thresholds governed by the presence of 10 meter-scale boulders in the channel. We thus expect river segments that are influenced by GLOFs to erode more rapidly than rivers with similar geometry and characteristic grain size and lithology without GLOFs. Therefore, we expect that GLOF-influenced rivers will drive their non-GLOF influenced tributaries toward higher $k_{sn}$ for the same erosion rate than if runoff-driven floods were the dominant erosional mechanism. If correct, this effect should be detectable in the geometry of tributary channels (Figure 2D).

Secondly and similarly, we expect GLOF erosion may be associated with discrete steepened reaches (knickpoints) in tributary channels near their outlets into larger trunk streams. In our proposed model for GLOF erosion, knickpoints should form in tributaries as a result of pulses of GLOF incision in the trunk stream. A concentration of knickpoints near trunk streams where outburst floods are more frequent would support an erosion model where GLOFs are an important factor. This is not to suggest that outburst floods are the only means by which knickpoints can develop at confluences. Punctuated incision, which may result in steepened reaches developing in tributaries, has been documented at a variety of timescales in rivers with different characteristics, and knickpoints generated at regional base level may propagate upstream and stall at confluences (Gardner et al., 1987; Crosby and Whipple, 2006; Finnegan et al., 2014). However, we hypothesize that outburst flood-driven incision may be particularly effective at generating knickpoints at tributary junctions due to the magnitude of erosion that may occur in a single event, particularly for rivers where GLOFs are relatively frequent.

Thirdly, the removal of coarse sediment by GLOFs is expected to change river valley widths. We propose that outburst floods facilitate river incision by mobilizing very coarse sediment, including large boulders, that remains stationary even during large runoff-driven floods. The widths of valley floors should reflect the degree of aggradation at longer timescales than the width of the active channels (Schwanghart et al., 2016; Yanites et al., 2018). If floods clear out aggraded material, we expect to see a narrowing trend in rivers subject to more GLOF activity if our erosion model depicted in Figure 2D plays a substantial role in Himalayan river incision. To test this prediction, we analysed valley floor widths based on a discharge-adjusted normalized channel wideness index ($k_{wn}$, see Methods) to account for the typical power-law increase in valley width with discharge.

## 2 Methods

We evaluated the metrics described above for a region of central Nepal characterized by N-S trending rivers draining across the Himalayan range (Figure 1). These rivers differ significantly in the extent of upstream glaciated area at the LGM. To complete topographic analysis of this region, we used the Shuttle Radar Topography Mission (SRTM) 30-meter digital elevation model (DEM), patched with the Advanced Spaceborne Thermal Emission and Reflection Radiometer (ASTER) 30-meter DEM where voids exist in SRTM. For the knickpoint analysis, we supplemented the 30 meter SRTM DEM with the 10-meter resolution EarthDEM (Center, 2021). Since we were not concerned with resolving small-scale features in our $k_{sn}$ analysis and the EarthDEM contains more voids and artifacts than the SRTM, we determined that the lower resolution DEM was more appropriate for $k_{sn}$ applications. Topographic metrics were calculated using the TopoToolbox and Topographic Analysis Kit packages for Matlab, and the DEM was preprocessed to remove outliers and impose a minimum downstream gradient for analysis of channel profiles (Schwanghart and Scherler, 2014; Forte and Whipple, 2019). In this section, we provide further detail on the determination of these metrics.

### 2.1 Physical relationships in channel networks

In actively uplifting landscapes, the geometry of the land surface is governed by competition between uplift and gravity, mediated by a series of processes with a variety of controlling factors. In time, this competition tends to result in a time-invariant condition of topographic steady state (Whipple and Tucker, 1999; Willett and Brandon, 2002). For most of the Earth's surface, local boundary conditions for erosion are set by the pace of incision or aggradation associated with river channel processes. In channel networks, the relationship between channel slope and contributing drainage area can reveal the active erosional processes. Downstream reaches of the channel network, which are typically controlled by fluvial processes, are described by the power law function

$$E = K A^m S^n \tag{2}$$

where $E$ is erosion rate, $K$ is the erosion coefficient, which is governed by local lithology, climate, and the process that control incision in the area, $A$ is drainage area, $S$ is local slope, and $m$ and $n$ are empirical constants which have a range of possible values depending on local conditions. Under steady-state conditions, where uplift and erosion can be assumed to be equal,

$$S = (U/K)^{(1/n)} A^{(m/n)} \tag{3}$$

where $U$ is uplift (Whipple and Tucker, 1999). This equation can be recast as Equation 1, known as Flint's Law, where $k_s$ defines a channel steepness $(U/K)^{(1/n)}$. The parameter $\theta$, termed the concavity, which under conditions of spatially invariant uplift and erodibility equals m/n, represents the rate of change of channel slope with drainage area and is generally accepted to be insensitive to uplift rate (Flint, 1974). $k_s$ varies with uplift rate but contains units that are dependent on $\theta$. In order to make a reasonable comparison of $k_s$ among channels with different $\theta$, we must fix the value of $\theta$ to a reference concavity, $\theta_{ref}$, that represents an average value for the channels in the area of interest, typically between 0.35-0.65, although this value may vary widely depending on local factors (Wobus et al., 2006a).

## 2.2 Normalized channel steepness index ($k_{sn}$)

Fixing $\theta$ to $\theta_{ref}$ results in the normalized channel steepness index $k_{sn}$ which is calculated as a best fit value for a given channel reach and is frequently and effectively used as a proxy in broad comparisons of uplift and incision rates across landscapes (Wobus et al., 2006a). Equation 3 is recast as

$$S = k_{sn}A^{-\theta_{ref}} \tag{4}$$

We found a best-fit $\theta_{ref}$ of 0.3513 and used this value for all $k_{sn}$ calculations in this study. We used the Topographic Analysis Kit to calculate $k_{sn}$ using the "$k_{sn}$ChiBatch" function, which fits $k_{sn}$ for channel network segments using with the desired criteria (Forte and Whipple, 2019). To examine the tributary basin response to GLOF erosion in the trunk streams, we used basin-averaged $k_{sn}$ in our analyses. Equations 3 and 4 are derived from the detachment-limited stream power model (Howard, 1994), and a comparison of $k_{sn}$ among channels assume that all erode accordingly. To exclude channels that do not behave according to the detachment-limited stream power model, we set a minimum drainage area to define a stream as 2 km$^2$ to avoid most channels where debris flow action is the main erosion mechanism. We have also excluded tributaries where the trunk valley at the confluence point has geometry that is indicative of erosion by direct glacial action (U-shaped valleys), tributaries where the tributary basin was likely to have been glaciated in its headwaters at the LGM (and thus may have experienced GLOF erosion as well). We have included tributaries where the trunk channel has extensive headwaters on the Tibetan Plateau (data points above $10^{10}$ m$^2$ on Figure 3), although the extent of glaciation on the Tibetan Plateau is still debated and a wide range of possibilities may be realistic (Kirchner et al., 2011). If regions above 4200 meters on the plateau were potentially ice-free at the LGM, then our proxy for GLOF frequency (total drainage area above the LGM ELA) may not apply in these rivers. However, as those channels do not deviate from the overall trends we identified, we found it appropriate to include them.

Equations 3 and 4 are derived from the detachment-limited stream power model (Howard, 1994), and a comparison of $k_{sn}$ between channels assumes that both erode according to this model. Incision by lake outburst floods is a vastly more efficient process than incision by runoff-driven floods (Cook et al., 2018), in that it can do more erosive work on lower gradient channels with less contributing drainage area, meaning $k_{sn}$ analysis could systematically underestimate incision in channels in which outburst flooding is an important geomorphic agent.

## 2.3 Knickpoint distribution

For our analysis of knickpoint distribution, we used the "knickpointfinder" function in TopoToolbox to identify and inventory knickpoints in the study area (Schwanghart and Scherler, 2014) (Supplemental Figure S1). To reliably identify knickpoints which might be missed in the 30-meter SRTM DEM, we obtained the EarthDEM 10-meter DEM, which is itself downsampled from a 2-meter DEM derived from stereo pairs of optical satellite imagery (Center, 2021). Tributaries included in the knickpoint inventory are 1st-3rd order streams that drain into 4th or higher order trunk streams (see example river profiles in Supplemental Figure S2). Similar to our $k_{sn}$ analysis, we excluded tributaries to trunk streams that substantially drain the Tibetan Plateau since the extent of LGM glaciation on the plateau is much debated (Kirchner et al., 2011). We set a minimum relief of 20

meters as the threshold for inclusion, to minimize the possibility of false knickpoints arising from noise in the topographic data. Since knickpoints can arise from many different geologic processes, we conducted the knickpoint search on parts of the tributary network we assume to be most affected by potential geologically recent outburst floods in the trunk channel, within 2 kilometres of a trunk stream. We included all tributaries below the region we expect to have been modified directly by glacial erosion in the knickpoint search and binned them against upstream drainage area above the LGM ELA in the trunk stream. We also examined the knickpoints identified by the function to ensure that the vast majority of them represent bedrock features, rather than incision into valley fill (Supplemental Figure S3).

## 2.4   Normalized channel wideness index ($k_{wn}$)

Most fluvial networks are characterized by a power-law increase in the width of channels as a function of contributing drainage area (Leopold and Maddock, 1953). This relationship is governed by many factors, including erosion rate, lithology, and climate, among others. Particularly in regions where extreme events can generate massive sediment inputs, channel width increases with aggradation (Schwanghart et al., 2016). Unit stream power is greater where channels are narrower, so channels may narrow to more readily expose bedrock and facilitate incision (Croissant et al., 2017). Dynamic channel width may thus illustrate channel response to tectonic or process-driven forcing. Channel width follows a power law relationship with discharge (Leopold and Maddock, 1953), as

$$W = k_{w}Q^{b} \tag{5}$$

where $W$ is the channel width, $k_{w}$ is a channel wideness index analogous to $k_{s}$, while $Q$ represents (in our case, estimated) discharge from the water balance (mean annual precipitation, $P$ – evapotranspiration, $ET$) in each DEM cell. By fixing a best-fit reference value for $b$, we can examine local variation in channel wideness in response to enhanced erosion by increased GLOF activity (Allen et al., 2013; Yanites, 2018), using

$$W = k_{wn}Q^{b_{ref}^{*}} \tag{6}$$

To calculate $Q$, we estimated the contributing runoff from each DEM grid cell using mean annual precipitation $P$) from a 12-year (1998-2009) Tropical Rainfall Measuring Mission (TRMM) dataset (Bookhagen and Burbank, 2010; Bookhagen, 2013; Kummerow et al., 1998) and $ET$ from the Global Land Evaporation Amsterdam Model (GLEAM) (Martens et al., 2017) and used the resulting runoff estimate to weight cells when calculating contributing drainage area. TRMM data is broadly used in mountainous regions, including in the Himalaya (Bookhagen and Burbank, 2010). When compared to gauge data, TRMM can underestimate rainfall at the highest elevations (Bharti and Singh, 2015). GLEAM evapotranspiration data has been field-verified as broadly accurate across terrain types in a study of major watersheds in China (Ma et al., 2019). We found negative water balances in some regions of the Tibetan Plateau which are in the drainage basins of a few of the rivers included in our analysis. To address this, we assigned cells with a negative water balance to have a value equal to the lowest positive discharge value in the area.

To investigate the influence of GLOFs on channel width patterns, we used Google Earth imagery to make 1,598 width measurements from rivers across our study area, spacing measurements roughly equally along river reaches (Supplemental

Figure S1). We measured the widths of valley bottoms instead of the channels themselves, since the active channel can change in width rapidly with deposition from local landslides and subsequent evacuation of deposits. Since glaciers can extend far below the ELA and we aim to avoid analysing valleys subject to direct ice action, we avoided taking width measurements at elevations above 3,000 meters except in a few locations where a V-shaped valley profile was very well-developed. We determined the location of transitions from valley floors to hillslopes by observations of several features. Many valley bottoms

have riparian vegetation that is visually distinct from vegetation on the hillslopes. In parts of the study area where valleys and hillslopes are developed for agriculture, farm terraces rapidly narrow where the hillslopes begin to steepen, offering a simple visual indication of the base of the hillslopes. Fluvial terraces are visible in satellite imagery and aid in distinguishing active valley bottom from abandoned surfaces. We included terraces within 10m of the elevation of the active channel in the valley bottom measurements, since a single outburst flood may incise enough to remobilize terrace material several meters above the

active channel (Cook et al., 2018). Our assumption that the width of valley bottoms is analogous to the width of active channels is supported by the observed power law relationships between discharge and valley width in the field area. While the width of the active channel itself can vary significantly over a short time, we expect that, although individual large landslides or other events might cause localized aggradation, on aggregate over our study area the width of the valley floor should reflect longer-term trends given that the timescales inherent in significantly raising or lowering an entire valley floor (and thus widening or

narrowing it) should be orders of magnitude longer than timescales governing the width of the channel (Ray and Srivastava, 2010).

## 2.5 Statistical analyses

We calculated Spearman rank correlation coefficients (Spearman's $\rho$) and $P$-values using the Matlab "corr" function with the "Spearman" parameter. The Spearman's $\rho$ is a nonparametric measure of the strength of association between two variables,

specifically useful for testing for a monotonic relationship where the nature of that relationship is unknown (Spearman, 1987). $\rho$ is reported as a value between 1 and -1 indicating the strength of the positive or negative correlation. We chose the Spearman's test since it was unclear what functional form the expected relationships among our variables should take. We also used two-sample Kolmogorov-Smirnov (K-S) tests, which compare the empirical distribution functions of two samples (Massey, 1951). K-S tests were conducted and $P$-values calculated using the Matlab "kstest2" function. The piecewise polynomial smoothing spline shown in Figure 3 used to determine expected $k_{sn}$ at a given elevation was fit using the Matlab "cftool" utility in the

spline shown in Figure 3 used to determine expected $k_{sn}$ at a given elevation was fit using the Matlab "cftool" utility in the Curve Fitting Toolbox, with smoothing parameter $p$ = 4.4773e-09. We chose a spline fit as the relationship between elevation and $k_{sn}$ appears to be naturally piecemeal, with average $k_{sn}$ increasing nonlinearly with basin elevation until 2500 meters, at which point it begins to decrease (Figure 3B). In Figures 4D-4F, we used the "rhohat" function in TopoToolbox to determine the nonparametric dependence on upstream above-ELA drainage area in the trunk stream on the distribution of knickpoints,

compared to the overall distribution of tributaries. To produce these results, we used log drainage area as the covariate, and set the bandwidth to 2.

## 3 Results

### 3.1 GLOFs and normalized channel steepness ($k_{sn}$)

Along the course of the major Himalayan rivers, the mainstems typically drain glaciated areas, while many of the tributaries do not. We examined the relationship between channel steepness and potential glacial outburst flooding by calculating the average $k_{sn}$ in tributary basins which drain to rivers with upstream glaciers, specifically in cases where the tributaries themselves are unglaciated (Figure 1). If GLOFs are indeed a regionally important erosional agent, we expect that effective GLOF-driven erosion in trunk streams should drive a topographic response in their tributary basins, which do not have access to highly efficient GLOF erosion. Since $k_{sn}$ is strongly correlated with elevation in the Himalaya, we have accounted for the overall trend between elevation and average $k_{sn}$, and analysed basin averaged $k_{sn}$ in the context of deviation from the expected value at a basin's elevation. Overall, we find that rivers with a greater proportion of upstream glaciated terrain tend to have tributary basins that are generally steeper (Figure 3). We interpret this steepening of tributaries as being a response to accelerated incision rates in the trunk streams driven by GLOFs. Repeated GLOFs occurring from the same source areas along the same flow paths will produce a persistent difference in erosion rate between erosionally less efficient tributaries and GLOF-dominated trunk streams. This difference would require the tributaries that lack glaciated terrain to steepen to keep pace with erosion of the mainstem, leading them to steepen — as we observe (Figure 2D). One potential complication is that in small, very steep catchments, such as some of the tributaries examined in this study, debris flows can control channel geometry at drainage areas of up to several square kilometres (Dahlquist and West, 2019). Since channels incising due to debris flow action do not follow a power law relationship between slope and drainage area, the use of $k_{sn}$ as a simple uplift-incision proxy in these catchments is problematic (Stock and Dietrich, 2006). If debris flow erosion is indeed an important control on channel geometry along some of the 1$^{st}$ and 2$^{nd}$ order basins we studied, the steepening trend we observe in tributaries responding to more frequent GLOFs in the trunk channel may reflect steeper tributaries allowing for more frequent debris flows with longer runouts capable of doing more erosional work (Stock and Dietrich, 2006). Yet we argue that this additional erosional work still reflects steepness produced by incision of the main stem, i.e., via GLOF activity.

### 3.2 Knickpoint distribution and GLOF erosion

To verify whether patterns of knickpoints are consistent with GLOF incision being a prominent component of Himalayan erosion, we analysed the distribution of knickpoints on tributaries within 2 kilometres of 4$^{th}$ or higher order rivers (Supplemental Figure S1, Figure 4). In 3557 tributary channels, we found 3707 knickpoints with at least 20 meters of relief based on the 10-meter resolution EarthDEM. We log-binned knickpoint counts and total knickpoint relief by the amount of upstream drainage area above the ELA in the trunk stream that each tributary joins. We then assessed the proportion of knickpoints that are found in tributaries to rivers without glaciated headwaters, and we compared this proportion to that of tributary confluences in general. We found that knickpoints are less common in tributaries to rivers with no glaciated drainage area upstream (Figure 4). Only 37% of the knickpoints are found on tributaries to rivers without glaciated headwaters; in comparison, 51% of the tributaries analysed drain to rivers with no drainage area above the ELA. This effect is more pronounced when knickpoints are

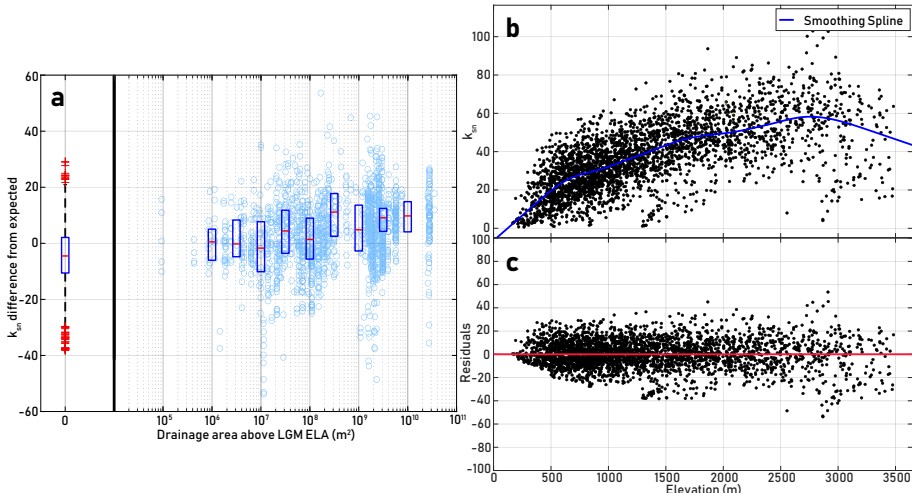

**Figure 3.** (a) Difference between tributary basin average $k_{sn}$ and expected $k_{sn}$ given basin elevation (based on residuals for spline fit shown in Figure 3b-c) versus total drainage area above LGM ELA in trunk stream basin (n = 3047). Box and whisker plot to the left of the break shows distribution of $k_{sn}$ differences for tributaries draining to rivers with no drainage area above 4200 meters. Box shows mean and upper and lower quartiles, whiskers represent 5th and 95th percentiles. Box plots to the right of the break show the mean and upper and lower quartiles for bins centered at boxes. Spearman's rank correlation coefficient (Spearman's $\rho$), which tests for a potentially nonlinear monotonic relationship, is $\rho = 0.3899$ with $P < 0.0001$ indicating a statistically significant positive correlation. We conducted a two-sample Kolmogorov-Smirnov test for the distributions of average tributary $k_{sn}$ of basins draining to channels with above-ELA drainage areas between $10^7$-$10^9$ m$^2$ (n = 754) and $10^9$-$10^{10}$ m$^2$ (n = 748) to determine if the samples come from significantly different distributions, and found the empirical CDF for the first group is larger with $P < 0.0001$.(b) Smoothing spline fit for tributary basin average $k_{sn}$ vs basin average elevation, calculated using the "cftool" utility from the Matlab Curve Fitting Toolbox. (c) Residuals for smoothing spline fit.

weighted by relief, with only  33% of the total knickpoint relief found on these tributaries to unglaciated rivers. In tributaries to rivers with substantially glaciated headwaters (draining at least 10 km$^2$ above 4200 meters) we find over-representation of the knickpoints, with  50% of knickpoints on these tributaries despite them making up only  41% of the analysed rivers. Again, this effect is accentuated when knickpoints are weighted by relief, with these knickpoints on tributaries to glaciated rivers representing  54% of the total knickpoint relief (Figure 4B). The greater proportions of knickpoints and total knickpoint

relief in the tributaries that drain into more glaciated channels support our conceptual model, wherein GLOF erosion can create knickpoints in tributaries at their confluences with the path of repeated outburst floods. These tributary knickpoints may stall at the confluences (Crosby et al., 2007; Goode and Burbank, 2009), or they may propagate upstream. By identifying knickpoints found up to 2 kilometres upstream from a potential GLOF path, we include both possibilities. We limited our analysis to the first 2 kilometres along the tributaries to minimize the possibility of crossing structural or lithologic gradients, which would risk

the inclusion of knickpoints formed by other conditions. Another point of interest in our knickpoint inventory is the difference between the distributions of knickpoints with respect to upstream glaciation between rivers above and below the physiographic transition (PT) (Figure 1). Only above the PT do we find a statistically significant offset between the distribution of knickpoints

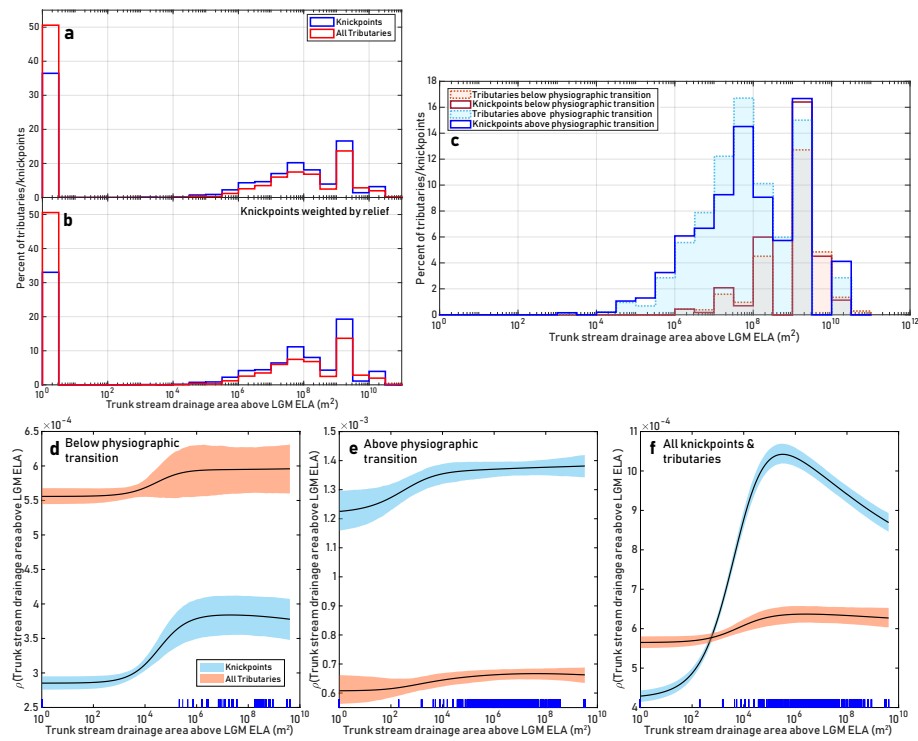

**Figure 4.** (a) Distribution of knickpoints (n = 3707) and analysed 1st and 2nd order tributaries (n = 3557) to 3rd or higher order rivers with respect to the area of terrain above the ELA drained by the trunk stream. Knickpoints included in the analysis are located on a 1st or 2nd order tributary within 2 km of a confluence with a 3rd or higher order trunk stream. Area is log-binned, the lowest area bin contains only knickpoints and confluences where the trunk stream does not drain any terrain above the ELA. See Methods for criteria for identifying knickpoints. (b) Same as 4A, but knickpoints are weighted by their relief. For both the relief-weighted and non-weighted knickpoint distributions, we conducted two-sample Kolmogorov-Smirnov tests for the distributions of knickpoints versus confluences with respect to above-ELA drainage areas and found the empirical CDF for the confluences is larger with $P < 0.0001$. (c) Comparison of knickpoints and tributaries located above (n = 1472 tributaries, n = 2549 knickpoints and below (n = 2085 tributaries, n = 1152 knickpoints) the physiographic transition (PT) (Figure 1). Including only those knickpoints and tributaries that drained to trunk streams with drainage area above the LGM ELA, we conducted two-sample Kolmogorov-Smirnov tests for the distributions of knickpoints versus confluences with respect to above-ELA drainage area, and found that the empirical CDF for the confluences is larger with $P \approx 0.01$ above the PT, while for the knickpoints below the PT we could not reject the null hypothesis with 95% confidence that they belong to the same distribution. (d-f) Results from "rhohat" function in TopoToolbox (Schwanghart et al., 2021) which returns the nonparametric dependence of the distribution of point features along a river network on a covariate, which in our case is upstream drainage area above the ELA in the trunk stream. Colored regions represent the bootstrapped uncertainty intervals and ticks along the x-axis represent individual data points.

vs. tributaries with increasing glaciated drainage area. This offset is not observed below the PT (Figure 4C). Figures 4D-F show the nonparametric dependence of the distribution of knickpoints and tributaries along the channel network on drainage

area above the ELA in the trunk streams (Schwanghart et al., 2021). In all cases, we see increased knickpoint prevalence in tributaries to trunk streams with potential upstream glaciation, when measured against the distributions of tributaries overall.

### 3.3    Valley widths and the role of GLOFs in "clearing the pipeline" of sediment

We expect that variation in valley floor width reflects the extent of alluviation. Wider valleys should have less frequent bedrock exposure, reflecting aggradation and slower incision. Valleys on GLOF paths should be systemically narrower than expected for

a given discharge if GLOFs are clearing out sediment and driving rapid incision frequently enough to control river morphology. As described in the Methods, we measured the widths of valley floors and calculated a normalized wideness index, $k_{wn}$, adjusted for the expected power law increase in channel width with discharge incorporating the a discharge estimation to account for the considerable variation in precipitation throughout the study area (Allen et al., 2013). Measurements of valley width corroborate our inferences from $k_{sn}$ and knickpoint occurrence: we find distinct trends in the relationship between valley

width and discharge, with rivers that have upstream glaciers being narrower at lower discharges than rivers without glaciated headwaters (Figure 5A). Moreover, among rivers that do include glaciated terrain, valleys with more glaciated drainage area tend to have lower $k_{wn}$ (Figure 5D). These observations suggest that GLOFs keep valley bottoms free of coarse sediment that broadens valleys and armors the bedrock channel bed against erosion. In other words, more frequent GLOFs "clear the pipeline", preventing clogging and allowing valleys to remain narrow. This is not simply a binary relationship, i.e., we do not

see valleys with upstream glaciers relatively free of alluvium versus those without glaciers containing substantial fill, but rather find that the valley width appears to depend on the frequency or magnitude of the floods as inferred from upstream glaciated area (Figure 5D).

### 3.4    Influence of uplift and erosion on geomorphic metrics

Several other differences across the central Himalaya of Nepal are expected to influence river valley morphology, most notably

the pronounced south-to-north increase in uplift and denudation rates. Most of the differences we document as being related to GLOF activity are between different N-S trending river valleys, i.e., between rivers with glaciated headwater versus those without, so we do not expect that differences in uplift and erosion rates are a major confounding factor in our analysis. Nonetheless, to test whether and how differences in uplift and erosion might have complicated our analysis, we examined a subset of basins in the study area which have published estimates of uplift rate derived from river profile analysis (Lavé and Avouac, 2001) and

Beryllium-10-derived basin-averaged denudation rates (Godard et al., 2014). These basins contain no or minimal drainage area above the LGM ELA, ruling out the possibility of GLOFs as an important erosion mechanism within these basins and thus allowing us to test for the role of other factors. These basins vary by more than an order of magnitude in uplift and erosion rates, capturing much of the variation found in our study area as a whole (Supplementary Table 1).

    For the basins shown in Figure 6, we examined the deviation in $k_{sn}$ from that expected in tributaries to the main drainage,

applying the same methods as we used for the whole study area. Figure 7 shows $k_{sn}$ difference from expected plotted against uplift and denudation rates. We found no relationship between denudation or uplift rate and tributary $k_{sn}$.

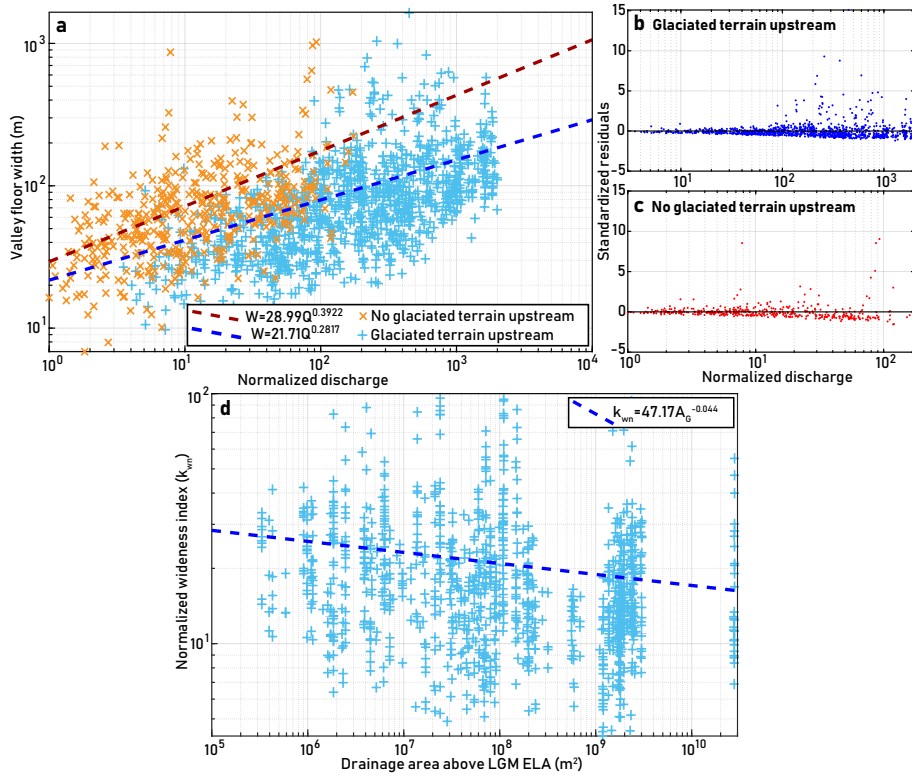

**Figure 5.** (a) Valley floor width versus discharge for rivers with and without headwaters above the LGM ELA, with power-law fits for valley wideness. Locations of valley width measurements are shown in Supplemental Figure 1. (b-c) Residuals plots for power-law fits shown in Figure 5A. (d) Normalized wideness ($k_{wn}$) versus contributing drainage area above the LGM ELA for valley width measurements in blue from Figure 5A. Here, $A_G$ refers to drainage area above the ELA. Spearman's $\rho$ = -0.2116 with $P < 0.0001$. We conducted a two-sample Kolmogorov-Smirnov test for the distributions of $k_{wn}$ ratios with above-ELA drainage areas between $10^7$-$10^8$ m$^2$ (n = 332) and $10^9$-$10^{10}$ m$^2$ (n=378) and found the empirical CDF for the latter group is larger with $P < 0.0001$. Fits shown here were calculated using the "nlinfit" function in Matlab.

We also took additional valley floor width measurements in the studied basins to test for the effect of denudation and uplift rates on valley width versus discharge trends. We fit width-discharge trends for all basins using Equation 6 (results shown in Figure 8). We calculated $k_{wn}$ for each basin, using a best-fit $b_{ref}$ of 0.3195. We found no correlation between width-discharge
trends and uplift and erosion rates.

Altogether, we observe no coherent relationships between between tributary $k_{sn}$ or basin $k_{wn}$ values and either uplift or denudation rate, suggesting that variations in these factors across the study region are not likely to explain the correlations we observe between our metrics of river morphology and the extent of glaciated headwater area. While our analysis based on spatial correlations cannot conclusively rule out other complicating lithologic, tectonic, and climatic factors, we have no reason
to expect these to produce the trends we observe.

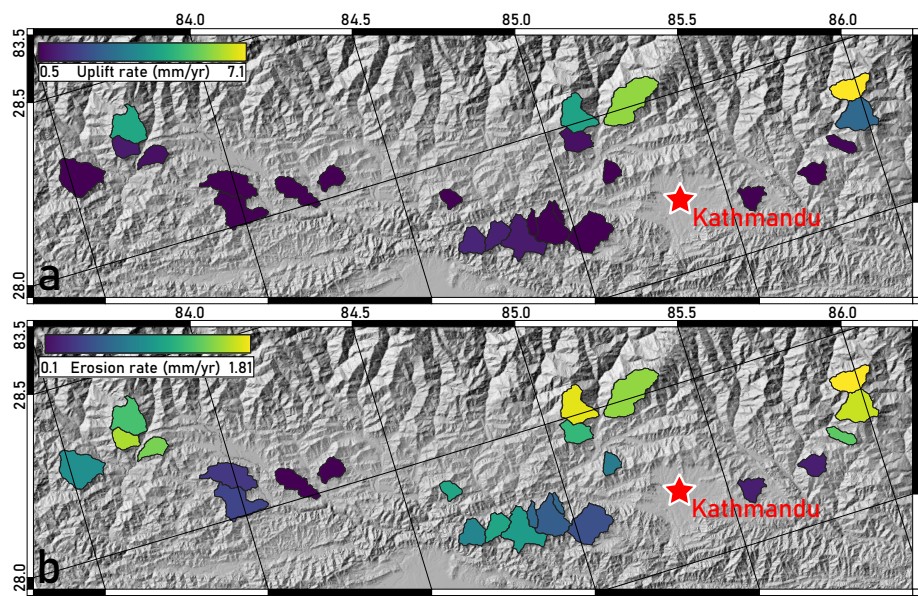

**Figure 6.** (a) Estimated uplift rate for basins with no or minimal drainage area above the LGM ELA (Lavé and Avouac, 2001). (b) Beryllium-10-derived basin-averaged denudation rates for the same basins (Godard et al., 2014).

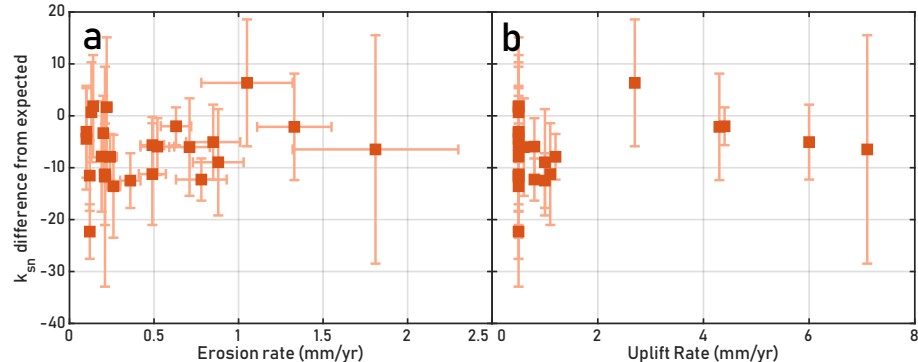

**Figure 7.** Difference from expected normalized steepness index in tributaries within basins shown in Figure 6 vs. erosion and uplift rates. Expected $k_{sn}$ at tributary elevation was calculated based on same spline fit shown in Figure 3. (a) $k_{sn}$ difference from expected in tributaries versus Beryllium-10-derived denudation rates (Godard et al., 2014)(. Spearman's $\rho$ = -0.0306 with $P$ = 0.5167. Vertical errorbars represent 1 standard deviation among tributary $k_{sn}$ values for each basin analysed. Horizontal bars are error reported from Godard et al. (2014). (b) Same $k_{sn}$ difference from expected in tributaries versus uplift rates (Lavé and Avouac, 2001). Spearman's $\rho$ = -0.0088 with $P$ = 0.8522.

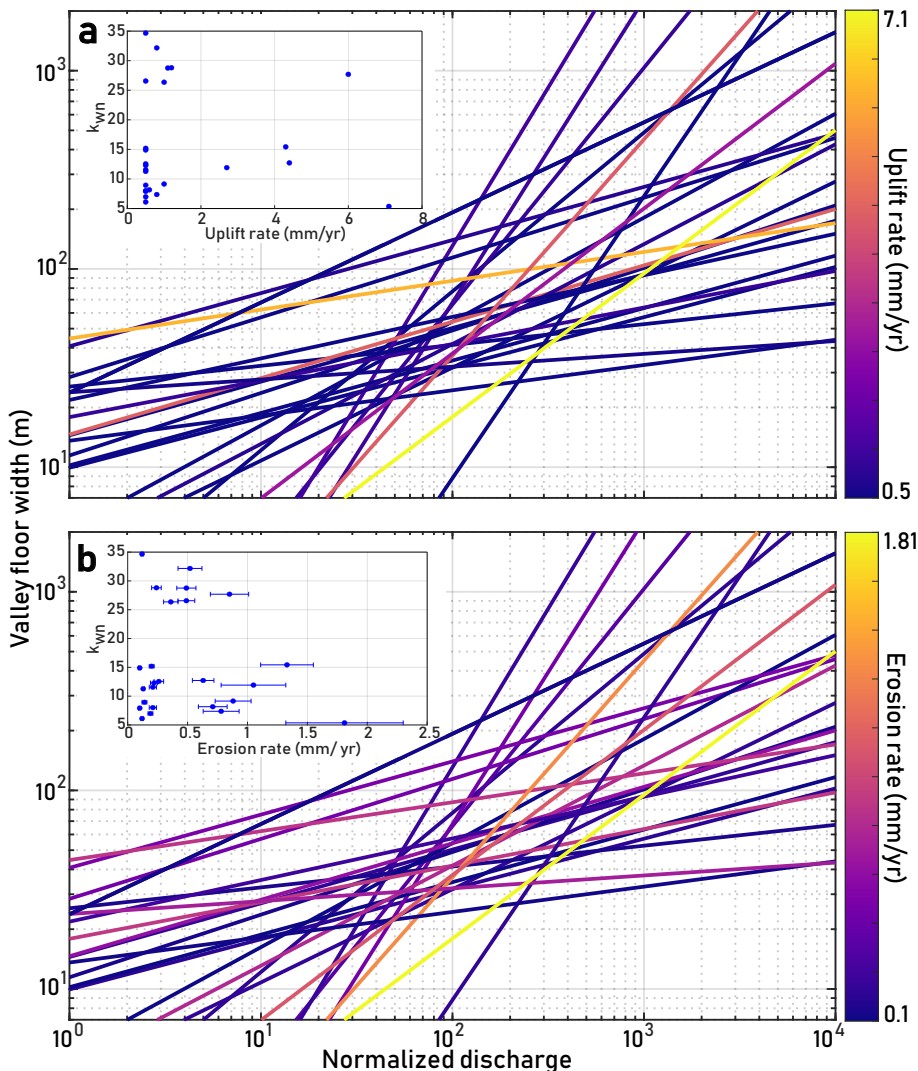

**Figure 8.** Best-fit lines for valley floor width vs. discharge for basins shown in Figure 6. (a) Lines show best-fit solutions for $k_w$ and $b$ in Equation 6 and are colored by estimated uplift rate for the basin (Lavé and Avouac, 2001)(. Spearman's $\rho = 0.2614$ with $P = 0.2069$ for $b$ vs. uplift rate. Inset: $k_{wn}$ versus uplift rate. Spearman's $\rho = 0.3257$ with $P = 0.3257$. (b) Lines show best-fit solutions for $k_w$ and $b$ in Equation 6 and are colored by Beryllium-10-derived denudation rate for the basin (Godard et al., 2014). Spearman's $\rho = 0.1289$ with $P = 0.5390$ for $b$ vs. denudation rate. Inset: $k_{wn}$ versus denudation rate. Spearman's $\rho = 0.0758$ with $P = 0.7187$.

## 4  Discussion

### 4.1  The Physiographic Transition: Shift from "top down" to "bottom up" erosion and complicating factors

Altogether, our analysis suggests that rivers in the central Himalaya bear characteristic signatures of erosion by glacial outburst floods, suggesting that these events are an important but largely under-recognized mechanism of regional incision. Yet GLOFs

can only be effective so far downstream. Cook et al. (2018) studied two major GLOFs in the Bhote Khosi valley, occurring in 1981 and 2016, and identified the location of rollover points along the downstream river profile where GLOF discharges attenuated to the point that a monsoon flood with the same recurrence would have greater discharge. Through the 20[th] and into the 21[st] centuries, the Bhote Khosi river has experienced GLOFs on a roughly 30-year recurrence (Mool, 1995), suggesting that the events studied by Cook et al. (2018) may be representative of typical GLOF-driven erosion in this watershed. These

rollover points lie very near the prominent physiographic transition (PT) that separates the precipitous High Himalaya from the gentler Middle Hills to the south (Figure 1). The abruptness of the PT reflects the topographic response to a steep gradient in uplift rate and is associated with a pronounced increase in erosion rates from south to north (Burbank et al., 2003; Wobus et al., 2006c; Godard et al., 2014). Over much of its length, the PT also represents a contact between the low-grade Lesser Himalayan Sequence and high-grade High Himalayan Sequence. In our study area, the Kathmandu Nappe juxtaposes High

Himalayan rocks into the Lesser Himalaya physiographic region, with no obvious effect on the local topography (Gansser, 1964), although lithology probably plays some role in controlling the PT. In many regions, lithology is an important control on landsliding and thus the formation of landslide-dammed lakes, and on the rate of breakdown of coarse sediment in channels by fluvial action, an alternative mechanism for the mobilization of coarse sediment without necessitating outburst floods. (Sternberg, 1875; Gerrard, 1994; Sklar and Dietrich, 2001; Dingle et al., 2017). However, for landslides triggered by the 2015

Gorkha Earthquake, Roback et al. (2018) observed no clear lithologic control, and landslide rates are likely considerably lower in the Middle Hills than the High Himalaya (West et al., 2015).

In this study, we found no significant difference in tributary steepness above and below the PT, though we did identify a subtle increase in potentially GLOF-driven knickpoint generation above the PT (Figure 4). Overall, the influence that GLOFs appear to have on channel steepness, valley width, and knickpoint generation, hint at the possibility of an erosional process

domain shift playing a role in the position and nature of major topographic transitions. Specifically, the PT may represent the position above which "top-down" GLOF-driven incision is prominent enough to maintain a persistent topographic signature. We identified a potential piece of evidence in the knickpoint inventory that hints at a possible supporting role for GLOFs in defining the PT by a process domain transition– since the extent of upstream glaciation seems to have a role in controlling knickpoint generation above the PT, but less so below it. This interpretation would be consistent with the locations of the

rollover points between GLOFs and monsoon floods in the Bhote Khosi.

A shift in process domain could also explain why we do not find distinct relationships between our topographic metrics and rates of uplift or erosion. However, we also recognize that these patterns may be due to the erosional role of landslide lake outburst floods (LLOFs), which can occur almost anywhere in the Himalayan river network, including the Middle Hills, and may be more frequent than GLOFs. For example, in the upper Sutlej River basin, 8 LLOFs occurred since 1973, comparable in peak discharge with the 1981 and 2016 Bhote Koshi floods, and the 2015 Gorkha earthquake triggered 25,000 landslides

forming at least 69 landslide dams (Collins and Jibson, 2015; Ruiz-Villanueva et al., 2017; Roback et al., 2018). If LLOFs are more frequent and widespread, but just as geomorphically effective, the signature of GLOFs may be subtle by comparison, although a top-down erosional regime might still be in force. The distribution of different types of landslide dams in space and time, more varied terrain in which they can occur, and the range of potential magnitudes makes adequate consideration of

LLOFs unfeasible in this study (Fan et al., 2020). However, we note that comparison of erosion rates over multiple timescales in this region of the Himalaya suggest a limited role for landslide-driven erosion in the Middle Hills (West et al., 2015) — suggesting that LLOF-driven erosion, like GLOF-driven erosion, may be less pronounced below the PT than above. Nonetheless, we recognize that the interplay of GLOF and LLOF processes is poorly constrained and we hope that this work serves as a starting point for inquiry into their effects as regional agents of erosion. Further complicating the topographic picture is the

fact that outburst floods are triggers of landslides along their paths, providing more opportunities for landslide lakes to form and ultimately drain in LLOFs. The interplay of different types of catastrophic floods and their aftermath makes it difficult to isolate the effect of GLOFs independent of other types of outburst flood. We expect that these inter-relationships may be responsible for much of the substantial scatter in our topographic data.

## 4.2    Implications for development of fluvial hanging valleys

Fluvial hanging valleys — steepened tributary reaches near their confluence with mainstem rivers — have been identified previously in the Himalaya and elsewhere (Wobus et al., 2006b; Goode and Burbank, 2009). While often considered enigmatic features, their persistence in the landscape has been explained by erosional mechanics that produce lower erosional efficiency in steeper river reaches with low sediment flux, during conditions of effective base level fall driven by rapid incision in mainstem channels (Crosby et al., 2007; Goode and Burbank, 2009). Coarse, frequently immobile landslide-derived sediment in

tributaries (Cook et al., 2018; Huber et al., 2020), coupled by GLOF-driven mainstem incision may contribute to conditions favoring fluvial hanging valleys without conflict with present theory of their formation. Furthermore, in our analysis of valley wideness we find that repeated GLOFs may inhibit trunk stream aggradation, which degrades fluvial hanging valleys (Goode and Burbank, 2009). Not all of the steepened zones near confluences that we have identified represent true hanging valley geometry, but our analysis of knickpoint prevalence in tributaries to glaciated rivers may suggest that repeated outburst floods in

a trunk stream may, under the correct conditions, control mainstem river incision and contribute to generating and maintaining fluvial hanging valleys. In this case, we explain the formation of these features as resulting from base-level fall in tributaries caused by rapid GLOF-driven incision of the mainstem. This produces persistent knickpoints at confluences where unglaciated tributaries enter trunk channels with upstream glaciation (Figures 2D-E), while processes that may maintain fluvial hanging valley geometry, such as infrequently mobile boulders in tributary channels and effective clearing of sediment in mainstems

are enhanced by GLOFs (Wobus et al., 2006b; Crosby et al., 2007; Goode and Burbank, 2009). We thus propose a connection between the formation of fluvial hanging valleys and upstream glaciation that leads to GLOF-driven erosion in the mainstem. Though the difference in knickpoint (and fluvial hanging valley by proxy) distribution that we observe associated with inferred GLOF activity is small, we expect many GLOF-associated tributary knickpoints to be smaller than would be picked up by the 10-meter DEM we have used. Future work might target analyses with higher resolution topographic data, perhaps over smaller

areas, to investigate smaller features.

## 4.3 Landscape Evolution from the Top Down

A simple end-member model of fluvial incision involves the formation of a knickpoint, or localized steepening, in response to uplift which manifests as a drop in a river's base level (Whipple and Tucker, 1999) (Figures 2A-C). In this model, increased steepness causes localized increases in erosion, and the knickpoint propagates upstream. Complexity in this process of incision and knickpoint propagation has been increasingly recognized: channels dominated by bedload abrasion may have knickpoint retreat rates that are decoupled from overall incision rates (Jansen et al., 2011; Wilson et al., 2013), and knickpoints may be smoothed out over years to decades in the presence of copious bedload and sufficient discharge (Cook et al., 2013).

Our analysis of Himalayan river channels suggests that "top down" incision driven by GLOFs may be another important factor in driving erosion and determining channel morphology in glaciated mountain belts. Based on relationships we have documented between the area of glaciated headwaters, tributary channel steepness, knickpoint occurrence, and valley widths, we propose that incision processes in the High Himalayan rivers of central Nepal are influenced in important ways from above, by outburst floods from the headwaters of the trunk streams. A critical controlling factor for the geometry of tributaries is their steepening in response to GLOF erosion.

If this process is as pervasive elsewhere as our data suggest it is in the central Himalaya, it would have significant implications for the evolution of orogens in response to tectonic and climatic forcing. In particular, an important role for GLOF erosion, such as that we have identified, implies that the relationship between tectonics and erosion may be modulated by the migration of the ELA. If uplift pushes terrain above the ELA, it could create new glaciers and glacial lakes that, in turn, accelerate GLOF-driven incision. This feedback, in tandem with the propagation of knickpoints from below, could link uplift and erosion rates in ways not captured in current models of landscape evolution. Alongside the effect of tectonics, climatic shifts can drive the ELA to higher or lower elevations, shifting dominant process domains and their signature relief structures to higher or lower elevations. Studies of landscape evolution and interpretations of river channel morphology and network geometry in mountainous environments should consider the influence of outburst floods as regional drivers of erosion, even where glaciers are no longer present. Altogether, our results suggest a rethinking is warranted of classic models of mountain river system evolution, to consider the role of glacial outburst floods as regional controls on erosion.

## 5 Conclusions

We found several lines of topographic evidence consistent with GLOF-controlled incision in rivers with glaciated headwaters in the Nepal Himalaya. Tributaries to GLOF-prone rivers are steeper than tributaries to non-glaciated rivers, and increasing extent of upstream glaciation in the trunk stream (and thus increasing GLOF frequency) increases this effect. We also found that the knickpoints are more numerous on tributaries to trunk streams with more glaciated terrain upstream, which provides further evidence for the steepening response that highly efficiently eroding outburst flood-dominated channels stimulate in their tributaries.

Additionally, rivers with glaciated headwaters have systematically narrower valleys than unglaciated rivers, indicating that GLOFs effectively sweep coarse alluvium from valleys and expose bedrock to erosion. This effect is increasingly prominent

with more upstream glaciation. Alongside previously reported evidence that outburst floods alone can mobilize the large boulders that frequently armor channels in major Himalayan rivers, the regional topographic analysis presented here suggests that GLOFs and other outburst floods, such as landslide lake outburst floods, may be a key erosional mechanism in these rivers. Our results point to a top-down model for valley incision in the Himalaya, in which erosion may be coupled to tectonics by uplift driving terrain above the ELA, expanding the reach of GLOFs, as opposed to (or in addition to) tectonically generated knickpoints propagating throughout Himalayan catchments from base level. GLOF-driven erosion may be important in other glaciated mountain ranges, appears to be independent of uplift and erosion rates, and should be considered in erosion models for such landscapes.

*Data availability.* Upon publication, the datasets generated and analysed during the current study will be made available in the Hydroshare repository, http://www.hydroshare.org/resource/2883cfeebb3a43f2b9a1b222e2cfff29

*Author contributions.* MPD and AJW conceived the study. MPD performed the analyses. MPD and AJW wrote the manuscript.

*Competing interests.* The authors declare that they have no conflict of interest.

*Acknowledgements.* We thank Kristen Cook, Jens Turowski, Georg Veh, and Missy Eppes for helpful discussions. We also thank William Medwedeff for the photograph used in Figure 2 and Wolfgang Schwanghart for editorial handling. John Jansen, Christoff Andermann and an anonymous reviewer contributed insightful comments that greatly improved this paper. Geospatial support for this work provided by the Polar Geospatial Center under NSF-OPP awards 1043681 and 1559691. This work was supported by NSF award EAR-1640894.

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
