# Peer review of "The imprint of erosion by glacial lake outburst floods in the topography of central Himalayan rivers"

_Earth Surface Dynamics, 2021_

## Author Response (AR1)

**Response to reviewers:**

We would like to thank both reviewers for their thoughtful commentary and apologize for the delay in returning the revisions. The reviewers have raised important points which we have addressed to the best of our abilities in the revised manuscript. We believe this paper is much improved for their insights.

**Reviewer 1:**

Normalized steepness index. In order to find a reference concavity, you need to restrict the analysis to settings where $(U/k)^{(1/n)}$ is constant in space, otherwise you are trying to fit a power law through data with multiple steepness indices. In the case of the Himalayas, it's well known that the trunk streams have huge convexities in them (Seeber and Gornitz, 1983; Lavé and Avouac, 2001). This is for good reason as the rivers traversing the Himalayas cross massive changes in rock erodibility and rock uplift rate. I *think* that your very, very low concavities could be a reflection of this issue. In other words, you are fitting concavities through many rivers that, as a whole, are not concave. In addition, although I applaud the use of discharge as a more realistic metric, this actually further obscures the issue because it's not clear how much of the atypical ksn scaling is driven by your integration of runoff. For theses reason, I would advocate one of two approaches in your revision

a) Assume a more typical reference concavity (e.g., 0.45) and redo the analysis with drainage area instead of discharge. In this way, you'd be assuming that the scaling here is the same as other locations (and indeed countless papers in the Himalayas have found concavities in this range).

b) Use your discharge, width, and slope measurements to just calculate mean unit stream power for the rivers (rho*g*Q*S/W). Then you can just look at patterns in river power directly, rather through the ksn scaling.

We thank the reviewer for asking us to dig deeper into these results. The low concavity was indeed a result of overfitting the convexities in the channel profile, and was accentuated using discharge instead of drainage area. This had given us erroneously low $k_{sn}$ values in high discharge channels. We've taken option (a), per the reviewer's suggestion, and redone the $k_{sn}$ calculation using drainage area and choosing more appropriate reference concavity (best fit for our study rivers) of 0.3513. Another potential problem with our analysis was the inclusion of potentially debris flow-dominated channels, a concern which the other reviewer raised. We have also adjusted the threshold drainage area for study channels to 2 $km^2$ to avoid these.

In the process of revisiting this analysis, we realized that using the $k_{sn}$ ratio between tributaries and trunk streams added some unnecessary complexity in our overall approach, especially in light of the concerns raised about potential complicating factors raised by both reviewers. For one, it introduces more potential error to focus on point measurements of $k_{sn}$ just near confluences (for example, because of the impact of local valley fill). Additionally, we determined that we did not adequately address the impact expected on mainstem $k_{sn}$ to confidently consider it in our analysis. Also, since we are arguing that GLOF-prone trunk streams adjust their width in response to repeated GLOFs, it seemed more prudent to consider

only one marker of channel geometry adjustment at a time. Instead, we have shifted to looking at the mean $k_{sn}$ of tributary basins independently, adjusting for the trend toward higher $k_{sn}$ at higher elevations. This also serves the purpose of diminishing the potential effect of short-term changes in valley floor elevation and width from local alluvation that 1) affect trunk streams more than tributaries, and 2) affect tributaries near confluences more than near the headwaters.

After making these changes, we still find a statistically significant increase in $k_{sn}$ in tributaries to increasingly glaciated trunk streams. We have added substantially to the discussion describing potential sources of the scatter in the data, most specifically the prevalence of landslide lake outburst floods as a similar erosional agent not limited to originating in glaciated areas.

Width scaling analysis.  This is a cool result, but one that could be an artifact of separating glaciated and non-glaciated rivers but not for the reasons assumed in the analysis (i.e. repeated GLOFs).  Tongues of ice could extend well below the 4200 m LGM ELA.  Indeed, today in the eastern syntaxis, which is the wettest area of the Himalaya, there are many glaciers that you can see in Google Earth that terminate at ~ 3,000 m in or near river canyons.  How do you know that the signal you are seeing is not being driven (at least in part) by the fact that you are looking at areas that were influenced directly by ice?  I know you've tried to avoid u-shaped valleys, but this is subjective.  Are there reconstructions of valley glaciers that you can appeal to to help have more confidence in this?  Even if there is no evidence for glaciers in these valleys, rivers that are downstream of glaciers are typically profoundly different than rivers that are not fed by glaciers.  In particular, rivers that are (or were) fed by glaciers are commonly braided (or were) and hence have wider valleys. In addition, glaciated rivers convey much more coarse sediment, and are therefore more prone to aggradation.  How can you be sure that the signal you are seeing is not simply an artifact of the difference in sediment supply between glaciated and non-glaciated basins?  Here, some photos from satellite imagery with interpretations could help to make your case.

We have been careful to avoid taking width measurements in valleys directly modified by ice. Only a tiny fraction of our width measurements were taken above 3,000 meters and for those we took at higher elevations, we confirmed a V-shaped valley profile. As mentioned by the reviewers, this evaluation is subjective, but we avoided any areas that were even slightly ambiguous for our higher elevation measurements. We have added additional clarifying text.

So, to clarify, our measurements were taken well below where glacial erosion would be a concern. The point the reviewer raises, though, highlights why these results are interesting. We found, contrary to the expectation that glaciers upstream would drive aggradation downstream, that upstream potentially glaciated terrain actually produced narrower valley floors than width-discharge trends would predict downstream — hence our argument that GLOFs serve to clear the pipeline of landslide (and glacial) derived coarse sediment. Our analyses mostly stopped many 10s of km below any modern glaciers, and closer to the source these trends might be different, but similar analysis at higher elevations would get complicated due to direct ice modification of the valleys as mentioned by the reviewer.

Hanging valley analysis. As pointed out by Crosby et al. (2007) and Wobus et al. (2006), this is an instability that is common in unglaciated places too. The difference in hanging valleys between your populations is not remarkable. A check on this analysis could be to look at the height of the hang and estimate what it implies about rates of LGM and later river incision? Is this plausible? I suspect that on a 10 m DEM, the signature of GLOF-triggered knickpoints would be tough to see.

The reviewer raises a valid point. To check our results and interpretation, we repeated our knickpoint analysis on the newly-available 10m-resolution EarthDEM (instead of the 30m SRTM DEM used previously; the EarthDEM product has only been released since our initial submission). The 10m EarthDEM is downsampled from a 2m product, and we tried to redo this analysis using the new 2m EarthDEM topography, but for the size of the study area the computational and memory requirements of topographic analysis (specifically DEM conditioning and flow routing) on such a high-resolution DEM proved unmanageable. In any case, the 10m EarthDEM should provide a more robust knickpoint inventory — and indeed the new results are consistent with our previous findings using the 30m SRTM data, suggesting that the outcome is not an artifact of DEM resolution. While it's true that the difference we observe is subtle, it is statistically significant and represents hundreds of excess knickpoints in the channels we have analyzed in aggregate.

For the other analyses, we continued to use the SRTM 30-meter, as we were not attempting to resolve fine-scale features, and the EarthDEM has holes and artifacts that make it unclear whether the improvement in resolution would mean an actual improvement in our $k_{sn}$ analysis.

In addition to performing the additional analysis with the higher-resolution DEM, we have rewritten the relevant section of text to soften our argument a bit regarding GLOFs serving as instigators for fluvial hanging valley formation. Rather, we argue that they are potentially important for generating knickpoints, which can in some cases stall and form fluvial hanging valleys.

Again, I love the goal of this paper. However, I am not convinced that the authors have isolated signals in the topography here that are actually diagnostic of the processes they are trying to study.

Thank you for the enthusiasm! We hope that the combination of improving the methods to remove the potential complicating factors and changes to the language and arguments has improved the manuscript and our conclusions are better supported for the changes.

A few minor edits below:

44- Should be "among" rather than "between"
66- no comma needed
127 - change "of" to "in"
139 - "stalemate" is a little colloquial here. Consider "steady-state"
205 - This is expected from hydraulic geometry scaling and hence you should cite Leopold and Maddock (1953) here

All of the preceding were changed and the Leopold and Maddock citation added.

**Reviewer 2:**

(1) Overtopping and collapse of landslide dams also causes gigantic floods (e.g. Fan et al. 2020, ESR 203). There is no obvious reason why such floods would be any smaller than GLOFs and certainly landslides are more common than moraines in these landscapes. Are landslide dam outbursts also more frequent? The authors do make brief mention of this other variety of flood (l.59-62) but perhaps there is more to be said. Where do 'LDOFs' fit in the proposed top-down valley incision model?

I raise this point because landsliding is considered the dominant agent of denudation/sediment flux in tectonically-active mountain belts. By undercutting hillslopes, GLOFs are likely to trigger landslides in a similar way to how we understand fluvial incision propagates from base level fall. I would guess that sudden GLOF-flushing of a valley fill may affect friction thresholds in hillslopes in ways even more likely to trigger failure. We know that lithology exerts a strong control on landsliding, and if the physiographic transition corresponds both to i) the chief contact between strong High Himalayan rocks and weaker Lesser Himalayan rocks, and ii) the transition in dominance of top-down vs bottom-up processes, then should we not expect that lithology is playing a role here too?

This is a good point and one that we had not adequately described in our initial submission. We limited our analysis to GLOFs because it seemed the simplest place to start examining the effectiveness of outburst floods as regional scale geomorphic agents. The distribution of GLOFs, though complicated in its own right, is at least predictable in that they can only originate in glaciated areas. Systematically analyzing the more stochastic landslide lake outburst floods is beyond what we can do in this study. However, we appreciate the reviewer's point and have added substantially to the discussion on these events and how they might complicate our results and add to the overall picture. We have also included a related discussion about lithology.

Hopefully, our analysis can serve as a starting point for future work on the specific impact of landslide outbursts. We have also added references to Fan et al. 2020 and Ruiz-Villanueva et al., 2017 as a more focused study of landslide floods in the Himalaya.

(2) The authors link alluviation to wider valley floors (higher k*wn) and narrowing to bedrock exposure. Fair enough. KPs are found to be more common in tributaries to rivers with glaciated headwaters, and this is interpreted as the propagation of base level fall triggered by GLOF incision along truck channels.

Mountain rivers characteristically undergo vertical fluctuations (of tens of metres or more) in valley floor elevation owing to downstream controls such as landslide dams or upstream sediment supply (see for instance Munack et al. 2016, QSR 149). Rivers with glaciated headwaters are subject to especially large fluctuations in sediment supply during a glacial cycle. While valleys floors were likely to have been alluviated during the last termination, the interglacial conditions of today typically promote incision along trunk streams (producing fill terraces) in response to the fall in paraglacial sediment supply. This drop in sediment supply is essentially a top-down process and the associated base level fall propagates up the tributaries as KPs incising old valley fills. Are the KPs shown in Fig. 4 cutting bedrock or valley fills? I recognise this may be difficult to determine in every case but it seems a key difference.

This is also a good point. After redoing the knickpoint inventory using higher resolution topographic data (see above), we also carefully reviewed the knickpoint locations in Google Earth and found they seem to overwhelmingly be in bedrock, especially for the above-PT group. There are a handful of areas where there appear to be some that have cut into valley fill, mostly in the Lesser Himalaya, but we estimate that these probably comprise less than 2% of the total population (for some of them it's difficult to determine). For the most part the knickpoints look to be in narrow, bedrock valleys. We suspect the 20-meter threshold for inclusion weeds out the vast majority of the knickpoints into valley fill. We have added some clarifying text to this effect in the main text, and some photos illustrating obvious bedrock knickpoints and some of the few apparent valley fill knickpoints to the supplement.

(3) I am still a bit unclear on why a threshold drainage area should apply to GLOF generation. Big GLOFs require big/tall moraines and glacigenic sediment volumes reflect glacial dynamics and supraglacial sediment loading (landsliding again!), neither of which is closely tied to area. What else could be responsible for this ~10 km2 threshold? A signal of debris flows and their runouts? The authors are up-front about this potential complication (l.254-261) and they are in a good position to appreciate the issue given they have studied debris flows in the region.

Nevertheless, the 0.48 km2 drainage area cutoff used in this study (from Roback et al. 2018) takes the analysis into channels that may experience a lot of debris flow activity. Judging by the slope-area data for central Nepal (Roback et al. 2018), many of these channels are well in excess of the S > 0.03 widely regarded to be dominated by debris flow incision (e.g. Stock et al. 2005, GSAB 117). Perhaps the point just needs some additional bolstering (l.260-261).

After the $k_{sn}$ recalculation with drainage area in place of discharge and a more realistic theta, per the other reviewer's suggestion, this proposed threshold is not as obviously relevant and has been removed from the discussion. We have also switched to a 2 km$^2$ threshold drainage area for our analysis to remove channels dominated by a debris flow signal, which would underestimate steepness in the basins. While some debris flows do run out to greater drainage area channels, 2 km$^2$ is below the rollover into the fluvial domain. Adjusting this value led to a clearer trend in Figure 3.

(4) Gigantic immobile blocks emplaced in the channel via landslides are presumably subject to partial breakdown by plucking and abrasion during high magnitude monsoonal floods. It seems likely that some fraction of the less gigantic blocks will be mobilised thanks to this in situ destruction, and if the intervals between GLOFs is long then it may be significant. Given the troubling lack of nineteenth century references in this MS, in situ breakdown of clasts may be an opportunity to cite Sternberg (1875, Z f Bauwesen 25); see also Dingle et al. (2017, Nature 544).

We have included some additional discussion to this effect in the discussion (Section 4.3) as well as the suggested references.

23 - Perhaps acknowledge the process of in situ breakdown of blocks over time, as noted above.

Discussion added, as noted above.

24-26 - Please clarify this a little. Do you mean the transport capacity increases with the passage of the flood bore? This is likely to vary depending on the nature of the dam failure. Perhaps spell out the general properties: geomorphic work done depends on the bed shear stress, the viscosity of the flow, the resistance to erosion (critical shear stress for entrainment) and the duration of the flood.

Not so much the transport capacity of the flood as a whole, but the ability of the initial water bore to mobilize new sediment. We have added text to clarify.

31 - Scherler et al. (2014) also point out that glacial dams are likely to reform after each failure thereby causing multiple possibly annual or decadal GLOF events. Perhaps under some circumstances GLOFs are rather frequent? This may be worth a mention.

We added a clarifying sentence and reference in Section 4.3 describing the roughly 30-year recurrence for GLOFs in the Bhote Khosi specifically.

35 - How representative of long term bed condition are today's observations? As noted above, valley floor elevations can fluctuate over tens of metres (or more) in these settings and this vertical instability is likely to vary over different timescales in response to: (1) the mag-freq of rainfall-runoff floods, (2) stochastic inputs of sediment to the valley floor via landslides, much of that material being paraglacial in origin, (3) GLOFs, and (4) the interglacial-glacial cycles.

Chronometric evidence (OSL and Be-10) of fluctuating valley floor elevations is emerging too, e.g. Dosseto et al. (2018, QSR 197).

This is certainly an important thing to consider. We would argue that, although short-term variability of valley floors certainly adds some error to our results, the vast majority of our datapoints are not overly affected by local alluviation, per our earlier response to this reviewer's question about knickpoints and valley fill. Furthermore, we have adjusted our $k_{sn}$ analysis to avoid dependence on values measured at or near confluences, which should further diminish the effect of recent valley fill.

39 - Top down proglacial/paraglacial sediment load is already recognised as important for bedrock fluvial incision. Is 'glacially driven' erosion the right term here? How about glacially conditioned?

Changed

43 - Are these opportunities really unique?

Changed to "ideal"

56-57 - Are there other factors worth mentioning that affect the frequency of GLOFs? e.g. relief (~sed source), valley width and valley steepness (~lake volume), lithology, seismicity?

We have briefly expanded on these factors.

67-69 - What aspect of mass balance? Please clarify.

Briefly expanded this sentence.

73-74 - Other mechanism exist too; I read this more in terms of a hypothesis that is to be tested here.

Rephrased this sentence to better reflect this point and to succinctly lay out the aims of this paper.

76-77 - Perhaps add that base level exists on a number of spatial scales. Base level can be local, e.g. landslide dams impose a local base level on the reach upstream which aggrades in response, a fault bound mountain front, or sea level.

We have added some clarifying text here.

97 - Does a best fit reference concavity value apply equally well across fluvial and formerly glaciated valleys? I expect valley troughs that have hosted glaciers for a good fraction of the Pleistocene might be less steep.

We've recalculated $k_{sn}$ using a more accurate reference concavity, per the other reviewers' comments. The less-steep valleys that probably hosted glaciers during the Pleistocene have been excluded from our analysis as much as possible (including from our reference concavity fit) so hopefully this shouldn't affect our results.

Fig. 2d - How is this a steady state ksn pattern? KPs are retreating upstream, hence the river profile is transient at the relevant timescale.

We changed the language of the caption to reflect that this is not actually a steady-state pattern.

101-106 - As noted above, perhaps be more specific here: entrainment/erosion depends on bed shear stress, a function of bed slope and flow depth.

We have added some additional language to this effect in both locations.

109 - All other factors being equal? What are these factors and what should we expect if they are not equal? Perhaps just rephrase.
110 - 'GLOF-influenced rivers will require lower ksn for the same erosion rate'. Is that the best way to express it? Is it more the case that GLOF-influenced rivers will tend to have lower ksn for a given erosion rate?

We reworded this section to clarify in response to these comments, and to reflect the change that we are no longer considering $k_{sn}$ in the trunk streams in the first place.

119-20 - If these KPs are solely GLOF induced, where are the base level induced KPs? Base level KPs are also known to cluster upstream of trib junctions due to the step down in drainage area (Crosby and Whipple 2006, Geomorph 82). Can the two KP types be differentiated? Presumably they amalgamate where they meet strong rock units.

This is something we would like to examine more closely in the future, but as of now it is not clear how to tackle it. Without trying to separate out tectonic knickpoints or those with any other origin, we are just looking at whether there are proportionally more where GLOFs may play a role, not assuming that they are the only, or even the most prolific, generator of knickpoints. We have added additional explanatory text to this effect.

139 - Stalemate is a good word but perhaps not the best term for topographic steady state. The processes are still active, it's just the external forms that remain invariant.

Changed the wording here.

151-2 - The concavity parameter is defined under conditions of spatially uniform uplift and erodibility, but this landscape contains sizeable KPs (>20m) that must therefore represent transient conditions. How to reconcile with the choice of reference concavity?

$k_{sn}$ analysis is widely applied conditions of non-uniform U and E. In this case, we aren't assuming steady state conditions but are using $k_{sn}$ to identify regions of channel adjustment to focused erosion (by GLOFs). We have recalculated $k_{sn}$ from the previous version using a value for reference concavity that is a far better fit to the whole study area.

208 - Or, rather it is unit stream power that increases with channel narrowing (seeing as W is the denominator).

Reworded this for clarity.

210 - Rather than framing the width-area relation in terms of the slope-area equation, I suggest this power law W-Q relation (among the other hydraulic geometry relations) should be attributed to Leopold and Maddock (1953, USGS PP 252).

Reworded and cited Leopold and Maddock here.

229 - This may be largely true, but I expect that valley fill alluviation behind a landslide, for instance, could elevate the thalweg in a matter of a few years only.

We added a clarifying sentence in here that we mean on aggregate over the whole region.

Good style to cite how the curves were fitted, e.g. Fig. 5 caption. Others should follow this example.

Having got to the end of the MS I unexpectedly found myself thanked in the acknowledgements! I think this may have more to do with Dr West's good manners than anything I could have contributed at the time!

All the more reason after this very insightful review!

---

## Author Response (AR2)

Re.: review, Central Himalayan rivers record the topographic signature of erosion by glacial lake outburst floods, by Dahlquist and West

The manuscript submitted for publication to ESURF by Dahlquist and West investigates the role of glacial lake outburst floods GLOF's on river incision and landscape formation over longer time periods. GLOF's have been recognized for many years now for their tremendous hazard potential and destructive powers. However, the role of these events in landscape formation, river incision and sediment transport has been largely overlooked. The topic is very timely and well suited for publication in ESURF. The authors use three independent methods to identify the topographic signature of GLOF's in mountain landscapes, 1) adjusted normalized channel steepness index, 2) knickpoint distribution and 3) adjusted channel wideness. The results show that rivers draining from glaciated headwaters, thus potentially affected by GLOF's, leave a clear signature on the river channel network, which is a novel finding. The manuscript has now undergone several rounds of reviews and public comments and I acknowledge that the authors have incorporated all the review issues and trimmed the paper to the essential findings. In particular, the authors have cut out the discussion of a bottom-up vs top-down interpretations to only top-down which I think is fair and helps the manuscript to focus on the essential massage. I have only few comments and suggest the paper for publication with very minor revisions.

My main concern is the interpretation and discussion of hanging valleys that might be the result of a more efficient channel erosion due to GLOF's in the trunk rivers, leaving tributaries stranded behind not capable keeping up the incisional pace. The existence of hanging valleys, especially in the very deep incised High Himalayan Range, has been recognized already before (e.g. Goode and Burbank 2009). The interpretation was until now that these tributary knickpoints are formed by pulses of aggradation caused by intense erosion periods or sediment deposition due to climate change or other catastrophic events. Temporary aggradation sets the erosional base level for the tributary higher and leave them hanging once the sediments in the main channel have been cleared out. In this manuscript the interpretation is fundamentally different. I do agree with the interpretation here but it is one of at least tow possibilities. I suggest adding a paragraph to the discussion section on the formation of hanging valleys and possible interpretation.

We thank the reviewer for the supportive comments and the constructive suggestions. Line numbers and a few of the comments seem to relate to the initial version of this manuscript, which we have noted where appropriate.

Minor comments:

Title: the word "record" implies that particular events can be dated or allotted to the topographic signature. Consider to change to "imprint" or similar.

We made a change to the title as suggested.

Abstract: Contains no information about the three methods applied in this study, 1) ksn, 2) knickpoint distribution, 3) channel wideness. Consider to include a sentence on the applied methods.

We added clarification as to the methods applied in the abstract.

Line 24: Add Huber et al. to the citation

Added

Line 80: "from the bottom up" is this this wording still needed here or rather confusing?

We removed this specific wording and reworded the passage to clarify.

Line 90: Is this really poorly understood? The concept that the heavy tail of flood distributions (extreme events) is driving river incision has long been recognized.

We clarified that we mean the specific effect of outburst floods are poorly understood.

Section 2.1: What min area defines 1st order streams in your stream order classification?

2 km$^2$ - this is explicitly stated in section 2.2

Line 166: This is the TRMM 2B31 dataset, with 4km resolution compiled by Bookhagen and Burbank 2006 & 2010 to mean annual and mean monthly climatology layers. The citation is missing a web ref, journal, doi ….

This citation was specifically for the dataset and doesn't have a doi, though we have added a URL. We have added the Bookhagen and Burbank 2006 and 2010 references as well.

Line 165- 169 and 230 – 234: Has this been calibrated somehow? Should be not so difficult to calibrate with few longtime gauging station datasets. Is the GLEAM correction making any important changes? I understand it makes sense to correct for the orographic gradient of precipitation, however, often global evaporation datasets lead to a negative water balance when compared with TRMM data. Was this observed here?

We added a few references here where the TRMM and GLEAM datasets were verified with field data in similar environments to our field area. We also added clarification about how we dealt with negative water balances in some areas of Tibet. To this comment, the lines 165-169 are no longer relevant, as we changed our k$_{sn}$ analysis to remove the discharge calculation.

Line 170: Water storage variations will be considerable on a seasonal time scale, etc. Add annual variation in water storage … or cut it completely. Might be confusing because it does not matter for your time scale.

We cut this for clarity as suggested.

Line 198: Cite the statement

Added a reference to Kirchner et al. 2011

Section 3.2: Glaciated catchments are not well defined in the manuscript. How do you define this? Using secondary shape dataset, or everything above ELA is named glaciated …? Glacial cover was not determined in the manuscript. Could be easily done but is maybe a bit too much at this stage. Be careful with statements regarding glacial coverage and glaciated catchment if this has not been proper defined.

We have added a sentence in Section 1.2 to clarify that when we say "glaciated catchment" for the purposes of this study we specifically mean any catchment with some drainage area above the LGM ELA.

Line 345: cite statement.

We added a few references here

Discussions in general: Could elaborate more in detail on different processes producing hanging valleys.

We expanded Section 4.2 to include more detail on controls on fluvial hanging valleys, including additional references. We also clarified in this section that we are not necessarily arguing for a fundamentally different mechanism for fluvial hanging valley formation with GLOFs, but rather that in tributaries to mainstem channels with relatively frequent GLOFs, outburst flooding may produce favorable conditions for fluvial hanging valley formation by mechanisms that have been previously identified by others.

Fig. 4: The vertical dashed lines are strangely distributed. It seems not all tick marks have a dashed line associated. Same for the horizontal lines, why three in the upper panel and only one in the lower one? The difference between a and b is not very easy to depict, seems on the first order the same distribution with minor changes on the y-axis. Maybe there is a better way to support the findings?

This comment seems to refer to an older version of the figure, as we have removed the dashed lines from Figure 4 in the most recent version. We hope that the other changes to this figure and the addition of panel c have clarified the points we were trying to make in Figure 4.

We have also changed Figure 5 very slightly due to a typo in the original (the exponents in the equations in panel A were negative and should not have been).

---

## Author Response (AR3)

Author's response:

In this final version of the manuscript, we have just added a photo credit to William Medwedeff to the Figure 2 caption and panels D-F to Figure 4, in response to the editor's helpful suggestion, along with a reference to Schwanghart et al., 2021 - ESPL and some brief explanatory text. We are very grateful for the attention and patience to our manuscript!